# Chemical nucleases are a robust alternative for RNase H cleavage of human ribosomal RNA

Hagen Wesseling[1], Dennis Krug[2], Marvin Wehrheim[1], Michael W. Göbel[2], Stefanie Kaiser[1]*

1 Institute of Pharmaceutical Chemistry, Goethe-University Frankfurt, Frankfurt/M, Germany, 2 Institute for Organic Chemistry and Chemical Biology, Goethe-University Frankfurt, Frankfurt/M, Germany

☯ These authors contributed equally to the work

* stefanie.kaiser@pharmchem.uni-frankfurt.de

## Abstract

Besides the abundant ribosomal and transfer RNA transcripts (rRNA and tRNA, respectively), there are tens of thousands of long coding (mRNA) and non-coding transcripts (lncRNA) within each cell whose modification profiles have not yet been elucidated. One reason for this is that most mRNAs and lncRNAs are low abundant and their purification prior to direct modification analysis by mass spectrometry (LC-MS) is highly challenging. State-of-the-art mRNA purification protocols are based on poly(A) enrichment with subsequent rRNA depletion using either magnetic pulldown assays or RNase H. While these methods are well suited for RNA sequencing, where residual rRNA content can be acceptable, LC-MS analysis of mRNA requires samples with less than 1% rRNA and higher yields, making the existing methods close to unaffordable. Thus, a new principle for low-cost (pre)mRNAs and lncRNAs preparation from total RNA will be beneficial for LC-MS analysis but also sequencing approaches. Here, we show that the use of **ARRR (artificial ribosomal RNA remover:** conjugates of standard DNA probes and a small-molecule chemical nuclease) is suitable for rRNA cleavage. In addition, ARRR has a higher target specificity compared to *E. coli* RNase H using regular DNA probes and only limited off-target RNA degradation is observed with ARRR. In summary, we present a promising tool with high potential to remove overly abundant rRNA, which might be used for enrichment of lncRNAs and (pre)mRNAs for downstream sequencing and MS-based analysis.

## Introduction

RNA modifications are increasingly recognized as critical regulators of gene expression, influencing various aspects of RNA metabolism, including stability, splicing, translation, and degradation [1]. Among these modifications, N6-methyladenosine (m6A), pseudouridine (Ψ), 5-methylcytosine (m5C), and 2'-O-methylation (Nm) have been extensively studied in messenger RNA (mRNA), as reviewed in [2,3].

Several techniques have been developed to map and quantify RNA modifications across the transcriptome. Sequencing-based methods have significantly advanced our understanding

**Data availability statement:** All relevant data are within the manuscript and its Supporting Information file.

**Funding:** 325871075-SFB 1309 to S.K.

**Competing interests:** The authors have declared that no competing interests exist.

of the distribution and potential functions of RNA modifications. For both indirect sequencing, where RNA is first reverse transcribed into cDNA, and direct sequencing by the Oxford nanopore technology (ONT), data robustness is firmly linked to the number of reads per transcript. This is an ongoing challenge as a closer examination of RNA class distribution by mass reveals that total cellular RNA comprises 80–90% rRNA and 10% tRNA, with the remaining 5% composed of (pre)mRNAs, lncRNAs and other RNA species. The estimated number of mRNA and lncRNA transcripts per mammalian cell ranges from $10^3$ to $10^5$ [4]. Given the existence of over 40,000 lncRNAs and around 30,000 distinct mRNA transcripts, this corresponds to only a few copies per cell [5]. Reaching enough reads for each individual mRNA or lncRNA in sequencing experiments remains a major challenge in the field. The high abundance of rRNA in combination with its homogeneity is the major reason for failed sequencing runs and therefore, stringent protocols for rRNA removal prior to sequencing experiments are needed [6,7].

The second major technology for analysis of RNA modifications is liquid chromatography coupled with tandem mass spectrometry (LC-MS/MS) [8]. This technique involves enzymatic digestion to release nucleosides, which are then separated on a column and analysed based on their $m/z$ ratios. The sensitivity of this method reaches the picomolar range (which equals injection of amols of an analyte), making it highly suitable for analysing low-abundant modifications. However, it requires highly pure samples, as any sequence context and thus the origin of the RNA modification is lost. Therefore, rRNA removal is even more important for MS analysis and needs to be more stringent than it is for sequencing. A variety of techniques and commercial kits have been established to cover the needs of rRNA removal and/or enrichment of mRNA.

Commercial kits for isolating mRNA typically use polythymidine DNA oligonucleotides (poly-dT) to hybridize with the poly(A) tails of mRNA (Fig 1A). Subsequent magnetic pulldown of the biotin-coupled DNA probes enables the enrichment of poly(A) RNA, while non-poly(A) tailed RNAs such as (pre)mRNAs, some lncRNAs, tRNA and rRNA are reduced. However, these kits may also co-purify other transcripts, such as polyadenylated rRNA or tRNA [9]. Consequently, these samples are referred to as poly(A) enriched.

Other kits use reverse-complementary DNA (rcDNA) to hybridize with rRNA and remove it *via* similar magnetic pulldowns (Fig 1B). Remnants of rRNA and tRNA present a bottleneck in this workflow [9,10]. Alternatively, there are kits for enzymatic digestion of RNA:DNA hybrids using RNase H (Fig 1C). In this approach, samples are treated with numerous rcDNA probes targeting rRNA, which is then substrate to RNase H and fragmented. By chemically altering the rcDNA probe, the specificity of RNase H can be greatly improved [11]. The resulting rRNA fragments are separated from the low abundant transcripts of interest, for example, by size [12]. Typically, multiple methods must be combined to obtain rRNA-free poly(A) RNA [13], but the high purity results in low single-digit ng yields. In a systematic assessment of rRNA abundance after multiple rounds of mRNA enrichment and rRNA depletion, it was found that the purity of samples increases after each step of the purification process [6]. But as a trade-off, the yield of the RNA preparation decreases with every round of purification. For sequencing experiments a lower number of reads is the consequence and for MS low-abundant modifications may fall below the limit of detection. In addition, rRNA depletion kits are highly pricy and handle a maximum of 1 µg of input RNA yielding single-digit nanograms of enriched mRNA at on average cost of 50 € per sample (and up to 200 € per sample depending on the supplier). Depending on the number of aliquots (to increase the final yield) and purification rounds, the usual cost per sample is currently around 200–500 €. One reason for the price is the need for specially modified

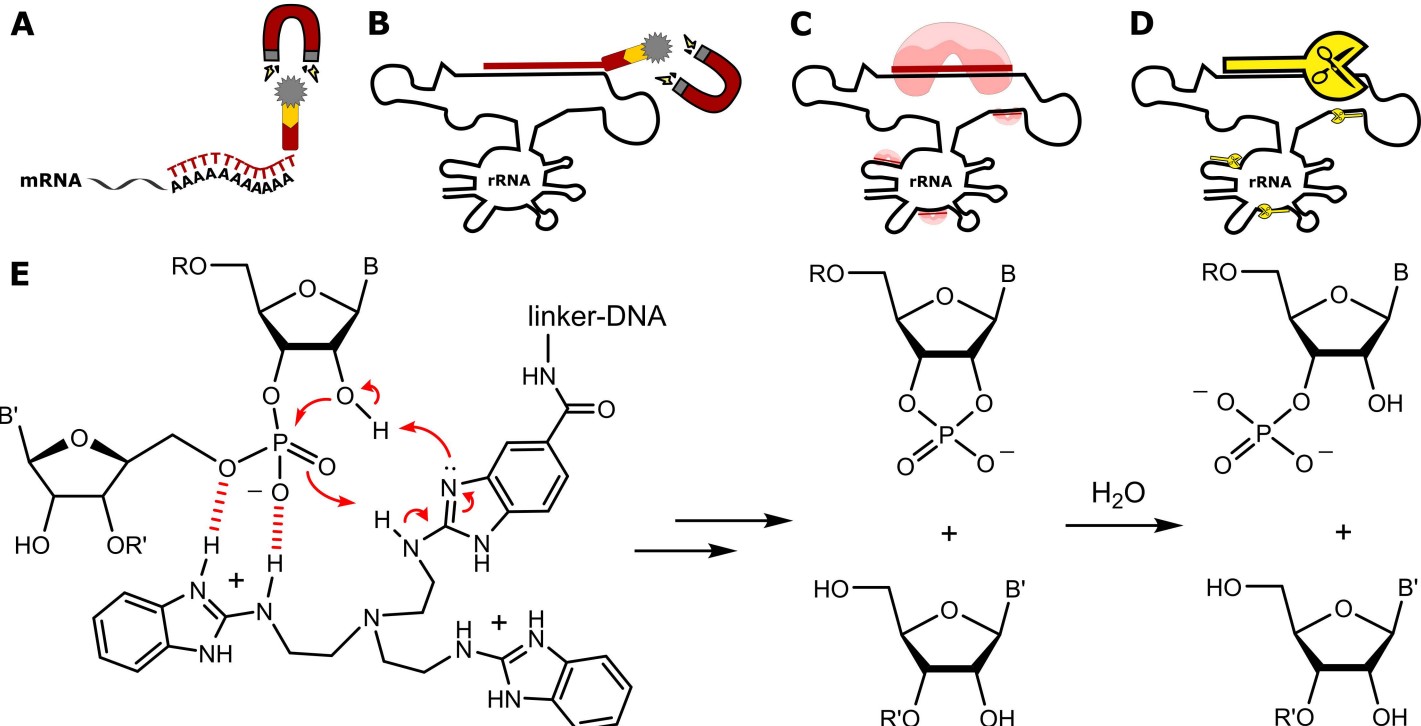

**Fig 1. Removal of rRNA from total RNA. A** principle of poly (A) enrichment. **B** rRNA depletion using magnetic pulldowns. **C** RNase H-based rRNA removal. **D** artificial ribosomal RNA remover (ARRR). **E** Cleavage mechanism of benzimidazole catalysts shown in 1D.

nucleic acids as probes to form the RNA:DNA hybrids that ensure a robust binding to the rRNA target and avoid unwanted off-target cleavages in RNase H based kits [11].

In this work, we want to challenge the RNase H-based approach for rRNA removal by using artificial nucleases (Fig 1D). We have previously reported on tris(2-aminobenzimidazoles) linked to DNA and their site-specific ability to cleave synthetic RNAs [14–16]. As illustrated in Fig 1E, the nuclease promotes attack of 2' hydroxy groups by general acid and base catalysis, which induces the hydrolysis of phosphodiester bonds within the RNA. Here, we have extended this strategy to biological samples, employing a cocktail of artificial nucleases that demonstrate both quantitative and sequence-specific cleavage of human long ribosomal RNAs. Importantly, we find no unspecific cleavage of non-target RNA using the artificial nucleases, while E. *coli* RNase H produces substantial unspecific cleavage for identical but non-cleaver modified DNA probes. Our data presents a cost-efficient and quantitative way for targeted RNA cleavage which will be useful for both sequencing and MS-based RNA analysis.

## Results

### Artificial Nucleases allow sequence specific rRNA cleavage

With our goal in mind that we want to cleave human rRNA, we isolated total RNA from HEK 293 cells. For initial method development, we used low amounts of total RNA that are well detectable by gel electrophoresis. 0.5 µg of the total RNA was hybridized with a DNA-probe of 19 nucleotides (nts) length with a 5'-cleaver (**1v** [15], and Fig 1E) or the corresponding unmodified DNA (**2v**). Both **1v** and **2v** are reverse complementary to human 28S rRNA. The RNA was incubated for 24 hours at 37°C in 50 mM TRIS buffer (pH 8) in absence or presence of 20-fold

molar excess of the DNA probes **1v** and **2v**. As a positive control, the reaction was performed using RNase H (note: RNase H incubation is only 20 minutes). The resulting cleavage products were monitored on an agarose gel (Fig 2A). We assumed that the total RNA might degrade under ARRR conditions due to the slightly alkaline pH and the long incubation period. Yet, as shown in Fig 2A, the total RNA remained intact in absence of DNA probes. Full size 28S rRNA has a length of 5025 nts and the DNA probe targets position 3789 (see Fig 2B). Thus, two products of 3789 nts and 1236 nts can be expected from site-specific cleavage. These products are observed for RNase H using both the conjugated (**1v**) and unconjugated DNA probes (**2v**). This indicates that the cleaver-conjugated DNA binds the target and does not interfere with RNase H cleavage. Under ARRR conditions a clear formation of the expected products is only observed in the presence of the cleaver-conjugated DNA probe **1v** but not in the presence of the unconjugated strand **2v**. This observation aligns well with our previous results [15]. In addition, we find that the 18S rRNA remained unaffected by the cleavage. From this experiment we conclude that ARRR DNA probes are suitable for sequence-specific cleavage of RNA and perform at least similar to RNase H.

We next designed and synthesized 23 additional artificial nucleases **1a**-**1x**, targeting either 28 S rRNA or 18S rRNA and purchased their corresponding unconjugated probes **2a**-**2x**. The cleavage position and expected fragments are summarized in S1 and S2 Tables. Each DNA probe was tested on total RNA and the cleavage products monitored on agarose gels. Probes **1a**-**1x** were used under ARRR conditions as in Fig 2A (S1 Fig) and in addition under RNaseH conditions (S2 Fig). Unconjugated probes **2a**-**2x** were only used under RNase H conditions (S3 Fig). For 19 ARRR probes (type **1**) we found successful cleavage under ARRR conditions, while 5 probes did not cleave as expected. Among the non-cleaving probes, 3 probes cleaved in the presence of RNase H, which indicates that for these 3 probes a hybridization occurred but no cleavage. This implies that the coupling of the cleaver to the DNA might have failed. But, according to our MALDI-TOF analyses, the ARRR inactive probes showed sufficient coupling (S4 and S5 Figs). Thus, we hypothesise that steric effects prevent the binding of the last three nucleotide hybrids so that the catalytic group does not reach the necessary proximity to the RNA. Of the remaining 2 probes, no cleavage was observed in the presence of RNase H, which indicates that the probe itself did not hybridize to the rRNA. This might occur due to the complex folding of rRNA. The structure of the cleavers implies that the cleavage reaction

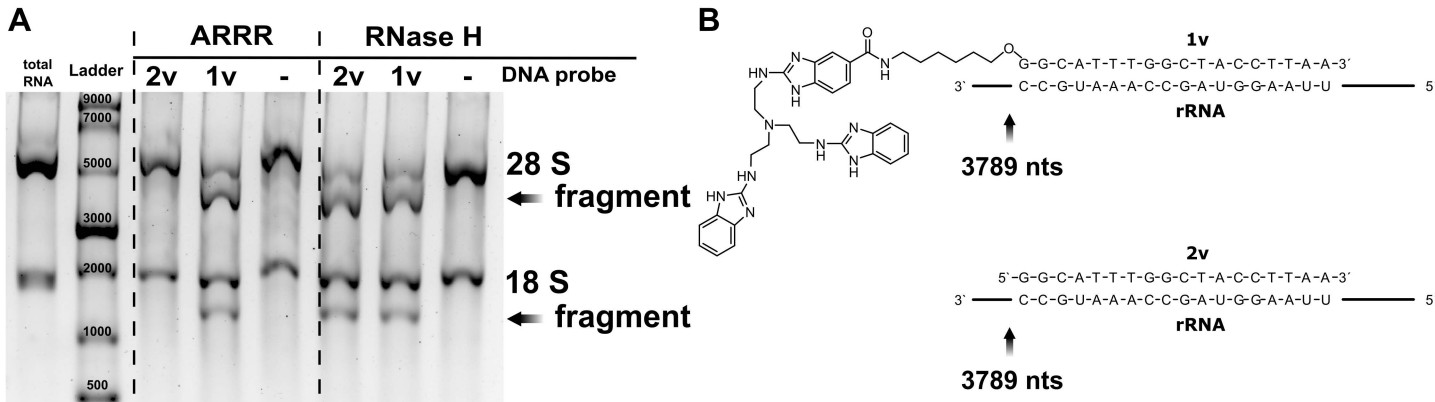

**Fig 2. The artificial nuclease (ARRR) cleaves 28 S rRNA into two fragments. A** Using ARRR, the DNA-catalyst conjugate **1v** splits the 28 S rRNA into fragments of approximately 4,000 and 1,200 nucleotides, in contrast to the unconjugated probe **2v**. RNase H digestion yields identical fragments with both probes. RNAs were separated on an 2% Agarose-TAE Gel. Stained with GelRed. **B** DNA probe **1v** and **2v** target sequence and cleaving site on the human 28 S rRNA.

is facilitated by divalent metal ions present in the solution. However, it is already known that the addition of up to 1 mM EDTA does not alter the cleavage activity [14]. This supports our hypothesis that steric reasons underlie the unsuccessful cleavages. An overview of the tested sequences and their cleavage site is summarized in S1 and S2 Tables.

## Optimization of reaction conditions

The objective of our method is to achieve sequence-specific cleavage of human rRNAs into shorter fragments for their size-based removal at high yields. From the 24 synthesized conjugates, 19 were active and mixed into an ARRR cocktail targeting both rRNAs. Calculations based on the cleavage sites predict cleavage products of less than 1000 nucleotides using this ARRR-cocktail. To improve quantitative cleavage of the ARRR-mixture, we systematically varied the reaction conditions to find the highest cleavage yield. First, we tested the effect of pH. Since the reaction is in part base-catalysed, acidic pH values are not suitable. Conversely, overly alkaline conditions could lead to unspecific base-catalysed hydrolysis of the phosphodiester backbone. We decided to test reaction conditions from pH 7.0 up to 8.6. Total RNA was denatured for 2 minutes at 95°C and then incubated with the ARRR-cocktail for 24 h at 37°C in 50 mM TRIS buffer. The products were again analysed on an agarose gel (Fig 3A). As expected, at neutral pH of 7.0–7.6, we observed larger fragments above 3000 nts, indicating insufficient cleavage. In the most alkaline environment (pH 8.6), unexpectedly short fragments were produced, suggesting nonspecific degradation. Consequently, pH 8 was selected for further experiments as here the cleavage yield and fragment size fit our expectations.

The artificial nucleases bring an alkaline micro-environment to the target RNA sequence which induces the cleavage of the phosphodiester. In the case of low abundance or even absence of the target RNA, unspecific cleavage of non-target RNA might occur. To prevent excessive artificial nucleases from causing nonspecific degradation, the stoichiometric ratios were optimized (Fig 3B). The ARRR-cocktail was diluted from 20 equivalents to 5, 1, 0.5 and 0.25 using 50 mM TRIS. As seen on the agarose gel in Fig 3B, 20 equivalents of each conjugate were necessary to achieve quantitative cleavage of both 28 S and 18S rRNA. Lower molar ratios resulted in longer products which reflect partly-cleaved rRNAs. For comparison with state-of-the-art methods, the same ARRR-cocktail was incubated with *E. coli* RNase H. Here, a 5-fold excess of probes was sufficient to achieve quantitative cleavage of rRNA. The difference in probe excess needed for cleavage can be explained by the design of artificial nucleases of type **1** for single turnover because they cleave at the end of the DNA-RNA duplex. RNase H, in contrast, cleaves in the center. The resulting shorter RNA fragments can dissociate from the DNA which is free to bind a new copy of RNA. In consequence, less DNA probe is sufficient for full substrate conversion. Apart from this advantage, we noted that the bands were smearier after RNase H and more defined after ARRR treatment.

Finally, the effect of temperature on the reaction was investigated (Fig 3C). We hypothesized that increased temperature might enhance the yield of ARRR cleavage. Again, we wanted to use RNase H as a positive control and thus we chose, in addition to *E. coli* RNase H, a thermostable RNase H, which should withstand the elevated temperatures. In addition, the thermostable RNase H is reported to be more specific than the *E. coli* RNase H [17] which was worth investigating after the smeary RNase H-derived bands in Fig 3B. Total RNA was incubated both under ARRR conditions and using RNase H at 37°C and 50°C using a slightly modified ARRR-cocktail. In this second cocktail the probes targeting 18S rRNA were omitted. Thus, we expect 18S rRNA to stay intact, unless unspecific cleavage by ARRR or the RNases occurs. Under ARRR conditions at 37 °C, the 28 S rRNA fragmented as expected and the 18S rRNA stayed intact as a sharp band. This is in harsh contrast to

the RNase H incubated samples. For both enzymes the 28 S rRNA was degraded, but in addition, the 18S rRNA degraded in the presence of RNase H from *E. coli* and remained largely intact in the presence of the thermostable RNase H. This second observation shows

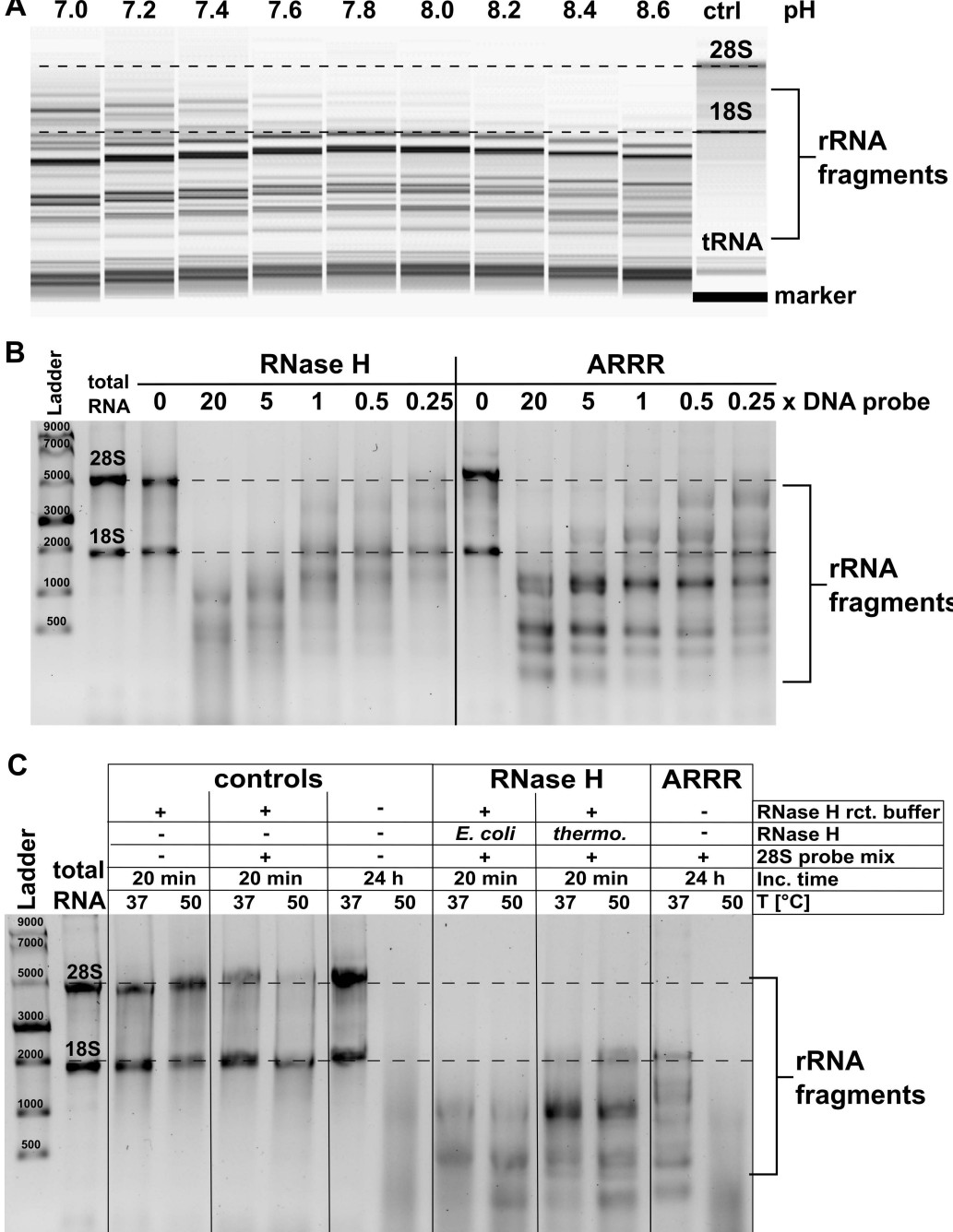

**Fig 3. Optimization of reaction conditions. A** total RNA was cleaved by ARRR with pH values ranging from 7–8.6 and then separated using a BioAnalyzer pico assay. **B** 2% Agarose gel of ribosomal RNA treated with various amounts of cleaver mix and cleaved by RNase H and ARRR. **C** 2% Agarose gel of RNA mixed with 20 eq. cleaver mix as described before, but the temperature was adjusted to 37°C or 50°C for all experiments.

*E. coli* RNase H has a clear tendency for unspecific RNA degradation with our DNA probe mixture. At 50°C, thermostable RNase H showed the best performance (judged as clearly defined product bands) while for *E. coli* RNase H and ARRR no clear product bands could be observed which implies unspecific degradation. Importantly, this degradation is also seen in absence of the 28 S probe mix, which indicates that the alkaline pH in combination with high temperature leads to RNA cleavage.

## Artificial nucleases show less nonspecific cleavage than RNase H

During our ARRR optimization experiments, we used RNase H as an established positive control. To our surprise, we found that RNase H showed unspecific cleavage of rRNA with our DNA-probe mixture. Unspecific cleavage might ultimately lead to a derichment of the transcripts of interest and thus lower signals. This would bias both MS analysis and sequencing studies. In Fig 4A, we compare the off-target cleavage effects of our ARRR method to those observed with standard RNase H assays. For this purpose, we mixed DNA probes targeting only 28 S rRNA (28 probe mix) or targeting both 28 S and 18S (28 + 18 probe mix). Since rRNA has many stable secondary structure elements, the release of the cleavage fragments in non-denaturing gels could be a problem. We therefore decided to repeat the experiment and additionally analyse it using denaturing chip gel electrophoresis (Fig 4B). For the 28 probe mix we expect to see an intact 18S rRNA band and the 28 S cleavage products. Indeed, for ARRR, the both gels meet our expectation. This is again in contrast to the *E. coli* RNase H, were we see a loss of the 18S band. For the thermostable RNase H, we find a similar intensity for the 18S rRNA as for the ARRR reaction. In the absence of the 28 S probe mix, no cleavage was observed, which indicates that RNase H was not overabundant and acted through its known DNA:RNA hybrid mechanism. This indicates that our 28 S specific probes bind the 18S rRNA and thus we performed a BlastN search of the 28 S probe mix against 18S rRNA. Indeed, we found that 8 probes had an overlap of 8 nucleotides each with 18S rRNA. We designed a new 28 S probe mix which omits these 8 probes (28 S v2 in Fig 4A and 4B) and here RNase H left the 18S rRNA uncleaved. We conclude that an 8 nts overlap is enough for *E.coli* RNaseH cleavage which must be taken into account when designing pure DNA probes for RNase H reactions. This is in contrast to the artificial nucleases where an internal 8 nts overlap does not lead to targeted rRNA hydrolysis. Yet, an overlap of 8 nts at the 5'-end and thus next to the cleaver group might lead to unspecific cleavage for ARRR and should be considered upon experiment design. As shown in Fig 4A and 4B, the probe mix 28 + 18 readily degrades both rRNAs as expected. Thus, we are confident that ARRR is a promising tool for the targeted cleavage of rRNA or other unwanted RNA prior to MS or sequencing analysis.

A bottleneck which had limited the access towards artificial nucleases in the past is related to the last step of the preparation. After assembly of the DNA strand by regular solid-phase synthesis, the catalyst phosphoramidite had to be attached in a manual coupling protocol [14]. This method, although effective, is more laborious compared to synthesizer-mediated conjugation. More recently we could develop improved protocols which allowed us to attach the cleaver directly on the synthesizer, albeit at the price of elevated amidite consumption [16]. Judging from a recent publication, 200 DNA probes were needed for the complete fragmentation of cellular rRNAs using RNase H [12]. We expect a similar number of probes to be necessary for the artificial nucleases. Thus, the question arises how clean the DNA probes must be after synthesis. Interestingly, we found that non-HPLC purified probes ("crude") performed as good as the purified probes (S6 Fig). This facilitates the preparation of the needed number of probes for complete rRNA fragmentation below 200 nts. Once the rRNA is smaller

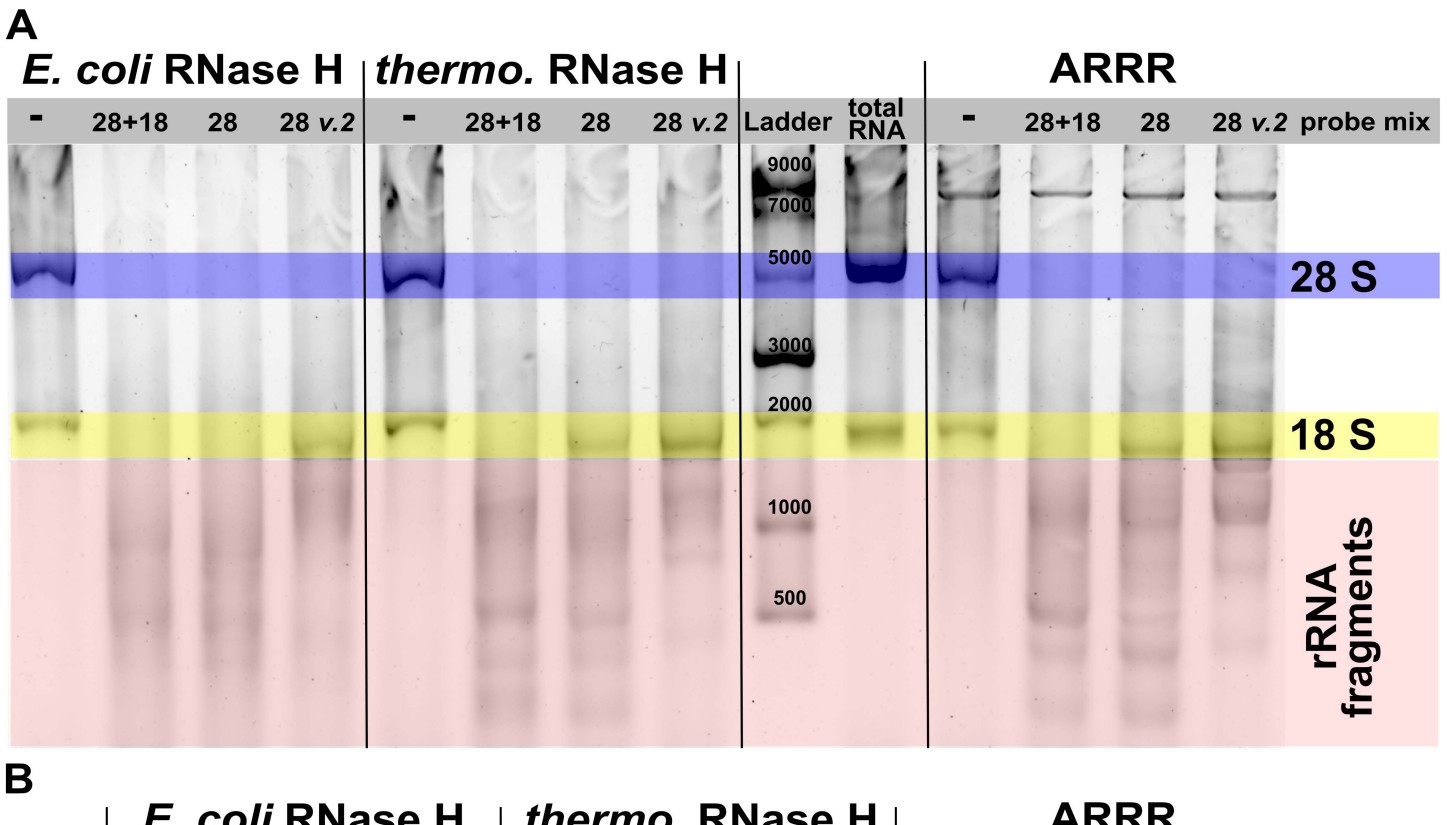

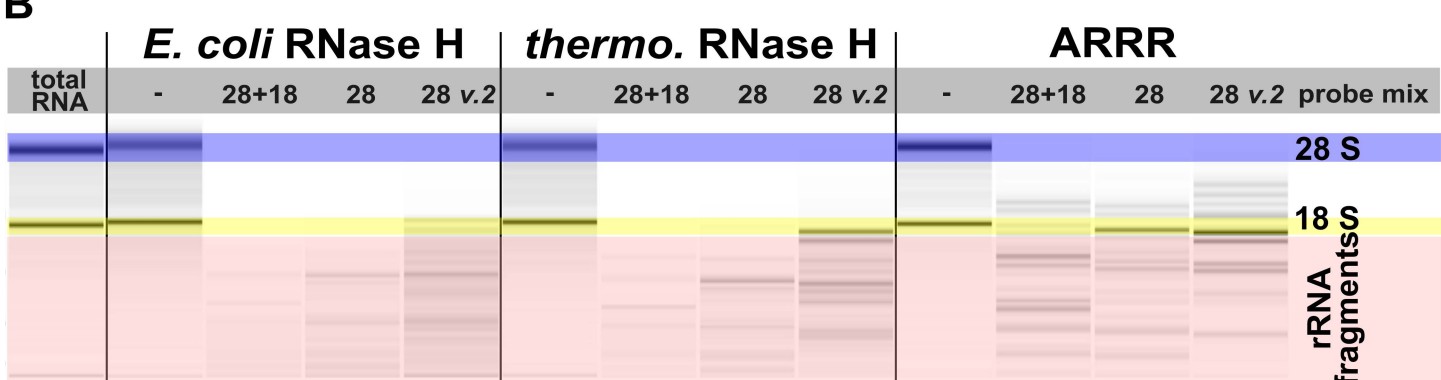

**Fig 4. Targeted cleavage of 28 S rRNA reveals that ARRR has little off-target effect. A** Total RNA was treated with an all rRNAs cleaving probe mix (28 + 18), a 28 S specific probe mix (28) or the 28 S specific mix where potential 18S targeting probes were excluded (28 v.2). RNAs were separated on an 2% Agarose-TAE gel and stained with GelRed. **B** The same experiment as in A was loaded on two Agilent BioAnalyzer pico chips for denaturing chip gel electrophoresis and merged into one graph.

than 200 nts, the rRNA fragments are removed from the total RNA alongside tRNA and other small RNA using small RNA isolation kits [18].

## Conclusion

From our previous work, we were familiar with the selectivity of artificial ribonucleases towards synthetically produced RNA [15]. In this work, we have expanded the substrate range of these conjugates towards native total RNA from human. We designed DNA-probes targeting rRNA, the most abundant (in mass%) RNA in human total RNA. Starting from our initial conditions published in 2019 [15], we tested both single probes and probe mixtures and we

successfully cleaved both 28 S and 18S rRNA into distinct fragments. The cleavage of rRNA is particularly interesting, as rRNA disturbs both sequencing and MS analyses and thus reliable tools for its removal will remain a major driver of future RNA analyses. This need is also reflected in the plethora of available kits for its removal. One option for the removal of rRNA is based on RNase H which cleaves RNA:DNA hybrids. The selectivity of this process strongly depends on the length and chemistry of the DNA probe. For best performance, unnatural modifications and or chimeric and locked nucleic acids are used as probes [11,19]. Our results also confirm the higher selectivity of the *T. thermophiles* RNase H compared to the *E. coli* RNase H [17]. This is in stark contrast to our artificial nucleases, that do not show unspecific cleavage, even under the prolonged incubation period of 24 hours at pH 8.0.

In summary, we could show that artificial nucleases are highly selective for their target sequence, even in the context of native human RNA. Therefore, we are convinced that artificial nucleases will become an alternative to existing rRNA removal kits. Thereby analysis of mRNA and lncRNA by either MS or sequencing will be more affordable and our understanding of the human epitranscriptome, or any organism's epitranscriptome, will grow.

## Materials and methods

### Chemicals

All salts and media were obtained from Sigma-Aldrich (Munich, Germany) unless stated otherwise. Unmodified DNA was purchased from Sigma-Aldrich and cleaver-linked DNA probes **1 a-k** were prepared as previously reported [15]. Probes **1 l-x** were not HPLC purified. Cleavage studies for probe **1 f** for an HPLC purified batch as well as for a crude product containing residual uncoupled DNA are shown in S6 Fig. The sequences, target sites and the compositions of the probe mixes are found in S1 and S2 Tables.

### Cell culture and RNA isolation

Medium for HEK 293T culture was DMEM D6546 high glucose supplemented with 0.584 g/L glutamine and 10% fetal bovine serum. Cells were grown at 37 °C in an 10% $CO_2$ atmosphere created by a CellXpert C170i (Eppendorf, Hamburg, Germany) and split every 2–3 days. After aspirating the medium, RNA was directly isolated from cell culture flasks using 1 mL TRI reagent and 200 μL chloroform according to the supplier´s protocol. RNA pellets were washed with 200 μL of 70% ethanol for 15 min at 12 000 xg at 4°C and re-suspended in ultrapure water, produced by a MilliQ EQ 7000 machine (Merck Millipore, Burlington, MA, USA). RNA concentration was determined on an N60 nanophotometer (Implen GmbH, Munich, Germany) using Lambert-Beer´s Law.

### ARRR Cleavage studies

Total RNA dissolved in ultrapure water was denatured at 95 °C for 2 min. For cleavage studies a solution containing 500 ng total RNA and 2.5 μL of 2.5 μM artificial nuclease were combined in an 0.2 mL reaction tube of 50 mM TRIS buffer (pH 8) to reach a final volume of 10 μL. The mixture was kept at 37°C for 24 h in a mastercycler X50 (Eppendorf, Hamburg, Germany) with a lid temperature of 42 °C to prevent condensation. The reaction was stopped by keeping the samples at -20°C.

### Agarose-TAE gel

For gel electrophoretic separation of total RNA and cleavage products 2% (w/v) agarose gel (Roth, #3810.5) was used. 500 ng RNA of each sample were mixed 1:1 with loading dye (NEB

#B0363S) and denatured at 50°C for 1 min. Agarose was dissolved in TAE buffer (40 mM Tris, 20 mM acetic acid and 1 mM EDTA in ultrapure water, brought to pH 8.3 with hydrochloric acid) which was also used as running buffer. Electrophoresis was performed for 2 h at 100 V in a running chamber (Roth) at 4°C. As size reference, a ssRNA Ladder (NEB #N0362S) was used. After separation RNA was stained with GelRed (Roth #223C.1) and visualized using a ChemiDoc MP Imager (Biorad, Hercules, CA, USA) in UV trans illumination mode (emission filter 590/110).

### RNase H digestion

For 500 ng total RNA 2.5 μL of the 2.5 μM probe were added and incubated at 95°C for 2 min. Afterwards samples were cooled down to 37°C for 5 min. 5 U RNase H and 1/10 (v/v) 10X RNase H reaction buffer (NEB, B0297S) were mixed to a total volume of 10 μL. The mixture was incubated for 20 min at 37 °C in case of *E. coli* RNase H (NEB, M0297L) and 50 °C for the *T. thermophiles* RNase H (NEB, M0523S).

### MALDI-TOF mass spectrometry of oligonucleotides

For each sample 0.5 μL of 3-HPA matrix (3-hydroxipicolinic acid saturated in $H_2O$:Acetonitrile (1:1; v:v)(Bruker Daltonics, Bremen, Germany), containing 10 mg/mL diammonium hydrogen citrate) are spotted on a ground steel target. The matrix solution is allowed to dry at room temperature. Then 0.5 μL of each sample (5 μM) are spotted on top of the dried matrix preparation spot. The sample is allowed to dry and the spectra are recorded in reflector and positive ion mode on an ultrafleXtreme MALDI-TOF-TOF mass spectrometer (Bruker Daltonics, Bremen, Germany).

### Supporting information

**S1 Fig. The artificial nucleases (ARRR) cleave human rRNA into two fragments.** Total RNA and the 5'-conjugated probes 1–24 corresponding cleavage product were monitored on a 2% Agarose Gel. Stained with GelRed.
(TIF)

**S2 Fig. RNase H digests also DNA:RNA hybrids containing 5'-conjugated probes.** Total RNA and the 5'-conjugated probes 1 – 24 corresponding digestion product were monitored on a 2% Agarose Gel. Stained with GelRed.
(TIF)

**S3 Fig. RNase H digests DNA:RNA hybrids with 5'-OH probes.** Total RNA and the 5'-OH probes 1 – 24 corresponding digestion product were monitored on a 2% Agarose Gel. Stained with GelRed.
(TIF)

**S4 Fig. MALDI/TOF-MS analysis of artificial nucleases 1a-1k.** Indicated peaks represent [M + H] + of the expected m/z of each probe 1a-k.
(TIF)

**S5 Fig. MALDI/TOF-MS analysis of artificial nucleases 1l-1x.** Indicated peaks represent [M + H] + of the expected m/z of each probe 1l-x.Not conjugated synthesis products were indicated as 2l-x.
(TIF)

**S6 Fig. Probe 1f cleaves 28 S rRNA as HPLC-purified product as well as crude synthetic product.** Total RNA was separated using a Agilent Bio Analyzer pico Assay.
(TIF)

**S1 Table. Sequence, cleavage position and resulting cleavage fragment lengths of 2 probes targeting H.s.** 18S rRNA. ** indicates the absence in the not 18S targeting probe mixes. (TIF)

**S2 Table. Sequence, cleavage position and resulting cleavage fragment lengths of 22 probes targeting H.s.** 28 S rRNA. * indicates the absence of the probe in the 28 S probe mix v.2 (TIF)

**S1 raw images. Original gel images.** (PDF)

## Acknowledgement

We thank Dr. Steffen Kaiser of the Campus Riedberg Mass spectrometry Service Facility for MALDI-TOF analysis of oligonucleotides.

## Author contributions

**Conceptualization:** Hagen Wesseling, Michael W. Göbel, Stefanie Kaiser.

**Data curation:** Hagen Wesseling, Marvin Wehrheim, Stefanie Kaiser.

**Formal analysis:** Hagen Wesseling, Dennis Krug, Marvin Wehrheim.

**Funding acquisition:** Michael W. Göbel, Stefanie Kaiser.

**Investigation:** Hagen Wesseling, Dennis Krug, Marvin Wehrheim, Michael W. Göbel, Stefanie Kaiser.

**Methodology:** Hagen Wesseling, Marvin Wehrheim, Michael W. Göbel.

**Project administration:** Michael W. Göbel, Stefanie Kaiser.

**Resources:** Michael W. Göbel, Stefanie Kaiser.

**Supervision:** Michael W. Göbel, Stefanie Kaiser.

**Validation:** Hagen Wesseling.

**Visualization:** Hagen Wesseling, Michael W. Göbel, Stefanie Kaiser.

**Writing – original draft:** Hagen Wesseling, Michael W. Göbel, Stefanie Kaiser.

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
