## [Decision Letter · Decision Letter 0]

28 Nov 2024

PONE-D-24-50983Artificial nucleases are a robust alternative for RNase H cleavage of human ribosomal RNAPLOS ONE

Dear Dr. Kaiser,

Thank you for submitting your manuscript to PLOS ONE. After careful consideration, we feel that it has merit but does not fully meet PLOS ONE’s publication criteria as it currently stands. Therefore, we invite you to submit a revised version of the manuscript that addresses the points raised during the review process.

The reviewers raise some concerns especially reagarding the reproducibility of the data and the cost comparisons.

We look forward to receiving your revised manuscript.

Kind regards,

Andreas Neueder

Academic Editor

PLOS ONE

“325871075-SFB 1309 to S.K.”

“This work was supported by the Deutsche Forschungsgemeinschaft [325871075-SFB 1309 to S.K.]. This project is funded by the European Regional Development Fund as part of the Union's response to the COVID-19 pandemic (EFRE-React). We thank Dr. Steffen Kaiser of the Campus Riedberg Mass spectrometry Service Facility for MALDI-TOF analysis of oligonucleotides.”

“325871075-SFB 1309 to S.K.”

5. We notice that your supplementary figures and tables are uploaded with the file type 'Figure'. Please amend the file type to 'Supporting Information'. Please ensure that each Supporting Information file has a legend listed in the manuscript after the references list.

Reviewers' comments:

Reviewer's Responses to Questions

**Comments to the Author**

1. Is the manuscript technically sound, and do the data support the conclusions?

Reviewer #1: Partly

Reviewer #2: Yes

2. Has the statistical analysis been performed appropriately and rigorously? 

Reviewer #1: N/A

Reviewer #2: N/A

3. Have the authors made all data underlying the findings in their manuscript fully available?

Reviewer #1: Yes

Reviewer #2: Yes

4. Is the manuscript presented in an intelligible fashion and written in standard English?

Reviewer #1: Yes

Reviewer #2: Yes

5. Review Comments to the Author

Reviewer #1: In this work, Wesseling et al. present the early stages of proof-of-concept work for the use of RNA-cleaving DNA probes in the depletion of rRNA for RNA-sequencinng and mass-spec applications. The idea has some merit, as shown by the limited studies presented here. The authors have not yet demonstrated its suitability for these tasks, but this is not a prerequisite for publication. I think the manuscript could be strengthened if the following questions/comments were considered:

1) Perhaps the title and abstract could be changed to better reflect the non–protein nature of the “artificial nucleases”. I did not realize that these nucleases were DNA probes with cleavage-inducing chemical groups until the end of the introduction. I was expecting some sort of engineered enzyme until I reached that point.

2) I am concerned by the use of native TAE gels for these analyses. The rRNA is extremely structured in nondenaturing buffers, even after being previously denatured, and this structure can hold the rRNA together even after cleavages have occurred. A denaturing gel or other denaturing method should ideally be used. Similarly, I wonder if the fuzzier bands in the RNase H treatment in 3B could be due to RNAseH remaining bound to the RNA and affecting its mobility.

3) Several commercial RNaseH based kits for RNA–seq exist at relatively low cost (from Illumina and NEB for example), as well as low–cost DIY methods (See Culviner et al, mBIO 2020). These methods are clearly suitable for basic RNA–seq, so the authors’ claim of high prices and that specialized modified oligos are required does not seem to be correct. However there may be more specialized cases where the off-target activity of RNaseH is a bigger issue (for example see Zinshteyn et al. RNA 2020).

4) In figure 4, what is the high molecular weight band above the 28S only in the ARRR lanes? Why does the supposed 18S band shift lower after ARRR treatment? Are the authors sure that this is not a 28S degradation product or a partially cleaved 18S?

5) Is there a roadmap for further optimizing the reactivity and specificity of these compounds? The requirement for 24-hour incubations in ph8 buffer seems like a risky and time-consuming requirement, and a serious impediment too widespread use

6) At several points in the conclusion, the word “prize” is used instead of price.

Reviewer #2: The authors proposed a new method for cleavage of human ribosomal RNA. The proposed method is promising and applicable. However, I have a few concerns.

Major concerns

1. This paper does not discuss the stability or reproducibility of their proposed system. I suggest adding some replicates and test the stability of their method.

2. The authors argued that one major drawback of the RNase H-based kits is its high price. How about the price of the newly proposed method?

Minor concerns

1. In the abstract, the authors claimed that they "find no unspecific cleavage of non-target RNA". But in the Result section, they only concluded "less unspecific cleavage" by their method (page 6). I think the claim in the abstract is too strong.

6. PLOS authors have the option to publish the peer review history of their article (what does this mean? ). If published, this will include your full peer review and any attached files.

**Do you want your identity to be public for this peer review?** For information about this choice, including consent withdrawal, please see our Privacy Policy .

Reviewer #1: No

Reviewer #2: No

---

## [Author Response · Author response to Decision Letter 1]

10 Jan 2025

Please see the detailed point-by-point response and cover letter.

---

## [Decision Letter · Decision Letter 1]

21 Jan 2025

Chemical nucleases are a robust alternative for RNase H cleavage of human ribosomal RNA

PONE-D-24-50983R1

Dear Dr. Kaiser,

We’re pleased to inform you that your manuscript has been judged scientifically suitable for publication and will be formally accepted for publication once it meets all outstanding technical requirements.

Kind regards,

Andreas Neueder

Academic Editor

PLOS ONE

Additional Editor Comments (optional):

Reviewers' comments:

Reviewer's Responses to Questions

**Comments to the Author**

1. If the authors have adequately addressed your comments raised in a previous round of review and you feel that this manuscript is now acceptable for publication, you may indicate that here to bypass the “Comments to the Author” section, enter your conflict of interest statement in the “Confidential to Editor” section, and submit your "Accept" recommendation.

Reviewer #1: All comments have been addressed

Reviewer #2: (No Response)

2. Is the manuscript technically sound, and do the data support the conclusions?

Reviewer #1: Yes

Reviewer #2: Yes

3. Has the statistical analysis been performed appropriately and rigorously? 

Reviewer #1: Yes

Reviewer #2: N/A

4. Have the authors made all data underlying the findings in their manuscript fully available?

Reviewer #1: Yes

Reviewer #2: Yes

5. Is the manuscript presented in an intelligible fashion and written in standard English?

Reviewer #1: Yes

Reviewer #2: Yes

6. Review Comments to the Author

Reviewer #1: (No Response)

Reviewer #2: The authors have addressed my comments in their response. Yet, I believe that the stability and the estimated price of their proposed system should also be included in the discussion section.

7. PLOS authors have the option to publish the peer review history of their article (what does this mean? ). If published, this will include your full peer review and any attached files.

**Do you want your identity to be public for this peer review?** For information about this choice, including consent withdrawal, please see our Privacy Policy .

Reviewer #1: No

Reviewer #2: No

---

## [Editor Report · Acceptance letter]

PONE-D-24-50983R1

PLOS ONE

Dear Dr. Kaiser,

I'm pleased to inform you that your manuscript has been deemed suitable for publication in PLOS ONE. Congratulations! Your manuscript is now being handed over to our production team.

Kind regards,

on behalf of

PD Dr. Andreas Neueder

Academic Editor

PLOS ONE